# Bacterial isolates, their antimicrobial susceptibility pattern, and associated factors of external ocular infections among patients attending eye clinic at Debre Markos Comprehensive Specialized Hospital, Northwest Ethiopia

**Zewodie Haile[1,2], Hylemariam Mihiretie Mengist[2]\*, Tebelay Dilnessa[2]\***

1 Department of Medical Microbiology Laboratory, Debre Markos Comprehensive Specialized Hospital, Debre Markos, Ethiopia, 2 Department of Medical Laboratory Sciences, College of Health Sciences, Debre Markos University, Debre Markos, Ethiopia

\* hylemariam@gmail.com (HMM); tebelay@gmail.com (TD)

## Abstract

### Background

External eye infection caused by bacteria can lead to reduced vision and blindness. Therefore, pathogen isolation and antimicrobial susceptibility testing are vital for the prevention and control of ocular diseases.

### Objective

The main aim of this study was to assess bacterial isolates, their antimicrobial susceptibility pattern, and associated factors of external ocular infection (EOI) among patients attended eye clinic at Debre Markos Comprehensive Specialized Hospital (DMCSH), Northwest Ethiopia.

### Methods

We conducted a cross-sectional study in patients with external ocular infections from January 1, 2021, to June 30, 2021, at DMCSH. Socio-demographic and clinical data were collected using semi-structured questionnaires. Following standard protocols, external ocular swabs were collected and inoculated onto blood agar, chocolate agar, MacConkey agar and mannitol salt agar (MSA). Finally, bacterial isolates were identified by Gram stain, colony morphology, and biochemical tests. Antimicrobial susceptibility testing was done by using the modified Kirby-Bauer disk diffusion technique according to Clinical and Laboratory Standards Institute (CLSI) guideline. Cleaned and coded data were entered into EpiData version 4.2 software and exported to Statistical Packages for Social Sciences (SPSS) version 22 for analysis. Bivariate logistic regression was applied to investigate the association between

**Data Availability Statement:** All relevant data are within the paper and its Supporting Information files.

**Funding:** The author(s) received no specific funding for this work.

**Competing interests:** The authors have declared that no competing interests exist.

**Abbreviations:** AST, Antimicrobial Susceptibility Testing; ATCC, American Type Culture Collection; CFU, Colony-Forming Unit; CLSI, Clinical and Laboratory Standards Institute; CONS, Coagulase-Negative Staphylococcus; DMCSH, Debre Markos Comprehensive Specialized Hospital; EUCAST, European Committee on Antimicrobial Susceptibility Testing; MDR, Multi-drug Resistance; EOI, External ocular infection; MRSA, Methicillin-resistant *Staphylococcus aureus*.

predictors and outcome variables. *P*-values $\leq 0.05$ with 95% confidence interval were considered statistically significant.

## Results

Two hundred seven study participants were enrolled in this study. More than half of them (57.5%, 119/207) were males, and 37.7% (78/207) of them were $\geq 65$ years old. A total of 130 (62.8%) bacterial isolates were identified, with Gram-positive bacteria accounting for 78.5% (102/130) of the isolates. *Staphylococcus aureus* was the most common isolate with a 46.2% (60/130) prevalence. Ciprofloxacin was comparatively effective against Gram-positive and Gram-negative bacteria. The prevalence of culture-confirmed bacteria was significantly associated with age groups 15–24 (AOR: 9.18, 95%CI: 1.01–82.80; *P* = 0.049) and 25–64 (AOR: 7.47, 95%CI: 1.06–52.31; *P* = 0.043). Being farmer (AOR: 5.33, 95% CI: 1.04–37.33; *P* = 0.045), previous history of eye surgery (AOR: 5.39, 95% CI: 1.66–17.48; *P* = 0.005), less frequency of face washing (AOR: 5.32, 95% CI: 1.31–7.23; *P* = 0.010) and face washing once a day (AOR: 3.07, 95% CI: 1.13–25.13; *P* = 0.035) were also significantly associated with the prevalence of culture-confirmed bacteria.

## Conclusion

The prevalence of culture-confirmed bacteria among patients with EOI was high in the study area. A considerable proportion of bacterial isolates exhibited mono and/or multi-drug resistance. Age (15–64 years), being farmer, previous history of eye surgery and less frequency of face washing were significantly associated with the prevalence of culture-confirmed bacteria. Bacterial isolation and antibiotic susceptibility testing should be routinely performed in the study area to combat the emergence of antibiotic resistance.

## Introduction

Understanding the health of the eyes is vital due to many factors. Several factors including, but not limited to, dust, high temperature, and microorganisms are factors associated with the occurrence of various eye diseases that can lead to blindness [1]. Besides, changes in the ocular microbiota are associated with ocular diseases [2]. Pathogenic microorganisms cause external ocular disease due to the virulence of microorganisms and the hosts' reduced resistance. Hosts' reduced resistance results from different factors like living conditions, socio-economic status, decreased immune status, chemotherapy, chronic diseases, and malnutrition. Bacteria are the major contributor to ocular infections worldwide [3]. The World Health Organization (WHO) recognizes corneal diseases as the major cause of vision loss second to cataracts, and an emerging cause of visual disability and blindness worldwide [4,5].

External ocular bacterial infections can cause a series of signs and symptoms such as the formation of pus, conjunctival hyperemia, lid edema, and even visual impairment. The causative bacteria may come from the outside environment or endogenously transported by blood. Normal flora can also cause infection, especially when they enter the aqueous humor or vitreous fluid [6]. Modification of normal flora in the conjunctiva and eyelid also contribute to ocular infections [7].

External ocular infection can be monomicrobial or polymicrobial. It is associated with many factors including contact lenses, trauma, surgery, age, dryness of the eye, and chronic nasolacrimal duct obstruction [8]. Some bacteria are part of the normal microbial flora in the conjunctiva and eyelids [9]. Bacterial infections contribute up to 74% of ocular infections globally. Studies reported that *Staphylococci* are the leading causes of external ocular infections worldwide [10] among Gram-positive bacteria, while *Pseudomonas aeruginosa*, *Klebsiella pneumoniae*, and *Escherichia coli* are the major Gram-negative bacteria isolated from external ocular infections [11].

The most common external ocular infections that may lead to blindness include conjunctivitis, blepharitis, dacryocystitis, keratitis, orbital, and periorbital cellulitis [11,12]. Conjunctivitis (red-eye) is the inflammation of the conjunctiva, and bacterial conjunctivitis could be characterized by the presence of mucopurulent discharge and conjunctival hyperemia [13]. Acute bacterial conjunctivitis is a common and a highly contagious infection in children and is usually treated empirically with broad-spectrum topical antibiotics [14].

Bacterial conjunctivitis is more common in young children and the elderly than in other age groups. The most common pathogens in bacterial conjunctivitis are *S. aureus* and *Streptococcus pneumoniae*. Other bacteria including *S. epidermidis*, viridans streptococci, *E. coli*, *P. aeruginosa*, and *Proteus mirabilis* had been isolated less frequently from bacterial conjunctivitis [15]. Keratitis is an inflammation of the cornea which may lead to corneal ulcer and corneal blindness [16,17] whereas blepharitis is an inflammation of the eyelid that can cause loss of eyelash. This infection may not remain localized and is known to spread to other anatomical sites of the eyes [11]. Additionally, dacryocystitis is also a clinical condition characterized by inflammation of the lacrimal sac, which usually occurs because of obstruction of the nasolacrimal duct [18,19]. It may also be related to a malformation of the tear duct, injury, eye infection, or trauma [20].

There is a globally high prevalence of antimicrobial-resistant *Staphylococcus* species among external ocular pathogens. Antimicrobial resistance to most groups of antimicrobials is increasing with a decline in the effectiveness of many commonly used topical antimicrobials [16]. The management of bacterial eye infections may involve treatment with broad-spectrum antibiotics. The use of broad-spectrum antibiotics leads to the development of resistance to the commonly prescribed drugs. The emergence of bacterial resistance towards topical antimicrobial agents may increase the risk of treatment failure [21]. Therefore, up-to-date information is essential for appropriate antimicrobial therapy and management of external ocular infections [22]. In Ethiopia, external ocular infections caused by bacteria are important public health problems [23,24]. However, there is a paucity of published data about the spectrum of bacteria and their antimicrobial susceptibility patterns among external ocular infected patients in the study area. Here, we hypothesized that the bacterial profile, their antimicrobial susceptibility pattern, and associated factors of external ocular infections in DMCSH could be the same with previous similar studies conducted in Ethiopia. This study assessed the bacterial causes of EOI, their antimicrobial susceptibility pattern, and associated factors among patients attended the eye clinic of DMCSH, Northwest Ethiopia.

## Materials and methods

### Study area and setting

The study was conducted among EOI suspected patients at DMCSH, which is found in Debre Markos town, the capital of East Gojjam zone in Amhara National Regional State, Northwest Ethiopia. DMCSH is the only tertiary hospital providing health care services for over four million inhabitants of East Gojjam and West Gojjam zones and the surrounding areas. In

addition, it is the only hospital with an independent tertiary eye clinic that provides both outpatient and inpatient services. All cases requiring tertiary care service in the area are referred to DMCSH. Besides, it has also a high patient flow as the eye clinic provides medical service for an average of 21,000 patients per year of which about 4,151 of them are clinically diagnosed as EOI. Moreover, it is the only hospital in the area providing bacterial culture and antimicrobial susceptibility testing services. Due to these reasons, DMCSH was selected as the only study site.

## Study design and period

A hospital-based cross-sectional study was conducted from January 01, 2021, to June 30, 2021.

## Source population

All patients who visited the eye clinic of DMCSH were the source population.

## Study population

All patients suspected of EOI visiting the eye clinic of DMCSH during the study period were the study populations.

## Eligibility

All EOI suspected patients except those taking and/or took antibiotics within the past two weeks and patients with acute physical eye injury were excluded from the study.

## Sample size determination and sampling technique

The sample size was determined by using a single population proportion formula considering 95% CI, 5% marginal error, and a 58.3% prevalence of EOI from a previous study conducted at Gondar Teaching Hospital, Northwest Ethiopia in 2017 [16].

$$n = pq \left( \frac{z\frac{\alpha}{2}}{d} \right)^2$$

Where n = sample size, p = 58.3%, q = 1-p = 0.417, d = margin of error that can be tolerated, 5% (0.05), Z = level of 95% confidence interval (1.96). Using the above formula, the sample size was calculated to be 376. However, the average daily flow rate of EOI in the eye clinic of the hospital was 15 and the study was conducted for 3 months with 22 working days each month making a total of 990 patients visiting the eye clinic. Since the total population size i.e., 990 was below 10000, we applied a sample size correction. Considering this, the final sample size was calculated as follows; N final = n/ (1+n/N) = 188. After considering 10% nonresponse, a total of 207 study participants were enrolled using a consecutive convenient sampling technique.

## Data collection

Demographic data (age, sex, monthly income, educational status, occupation, and residence) and ophthalmic clinical data (use of traditional medicine, history of eye trauma, history of eye surgery, frequency of face washing and presence of systemic disease) were collected using structured questionnaires, and physical examinations. Ocular specimen and interviewer-administered data were conducted by optometrist nurses after two days training. In addition, the clinical characteristics of patients were retrieved from patients' records. To identify the

clinical picture of EOI, all patients were examined using a slit lamp bio-microscope and diagnosed by an ophthalmologist. Then, the specimen was collected from each patient by gently swabbing the eye, the lower conjunctival sac, and lid margins using sterile cotton swabs moistened by saline. Specimen were aseptically obtained from EOI sites before the eye was cleaned with an antiseptic solution and antibiotic use [8]. In case of ulcerative blepharitis, lashes deposit, tear film foaming content and corneal punctuate erosions were swabbed. For dacryocystitis, only the pus was collected on a swab and inoculated onto a culture media. The collected specimens were directly transported to the microbiology laboratory of DMCSH for processing.

## Isolation and identification of bacteria

Gram staining was done from each sample for presumptive identification of gram positive and Gram-negative bacteria. The specimens were then inoculated onto blood agar (Oxoid Ltd, Basingstoke, UK) and chocolate agar and then subcultured on MacConkey agar (Oxoid Ltd) and mannitol salt agar (Oxoid Ltd) [9] for selective growth of Gram-negative and Staphylococci. All the inoculated culture media were incubated at 35–37°C for 24 hrs. Additionally, chocolate agar plates were incubated with a 5% $CO_2$ atmosphere. All culture plates were initially examined for growth after 24 hours and cultures with no growth were incubated for further 48 hours. For mixed colonies, a sub-culture on blood agar and chocolate agar was performed to get pure colonies. After obtaining pure colonies, further identification was done by using standard microbiological techniques including Gram stain, morphology characterization, and biochemical tests. Gram-positive cocci were identified by biochemical tests, catalase and coagulase positivity, optochin disk sensitivity, bile sensitivity and bacitracin sensitivity tests. Gram-negative bacteria were identified based on phenotypic characteristics and a series of biochemical tests such as carbohydrate utilization, indole production, mannitol fermentation, citrate utilization, lysine decarboxylation, H2S production, triple sugar iron utilization and motility testing [25].

## Antimicrobial susceptibility testing

Antimicrobial susceptibility testing was performed using the modified Kirby-Bauer disk-diffusion technique on Muller Hinton agar (MHA) supplemented with 5% defibrinated sheep blood for fastidious bacterial isolates (Oxoid Ltd) according to CLSI 2021 guideline [26]. Briefly, 3–5 colonies of the test organism were transferred into a tube containing 3 ml of nutrient broth/normal saline and mixed gently until the suspension becomes turbid and adjusted to 0.5 McFarland standards. Nutrient broth and normal saline were used to standardize the approximate number of bacteria with 0.5 McFarland standards in the Kirby-Bauer disk diffusion method. The suspension was swabbed uniformly onto MHA agar entirely by rotating the plate 60 degrees between streak for non-fastidious organisms, and MHA with defibrinated sterile 5% sheep blood for fastidious organisms. For Gram-positive bacteria, discs impregnated with antimicrobials ampicillin (10μg), chloramphenicol (30μg), gentamicin (10μg), tetracycline (30μg), trimethoprim/sulfamethoxazole (1.25/23.75μg), ciprofloxacin (30μg), ceftriaxone (30μg), clindamycin (2 μg), doxycycline (30 μg), erythromycin (15μg), penicillin (10U), and cefoxitin (30μg) (Oxoid Ltd) were used. The methicillin resistance pattern of *S. aureus* and coagulase negative staphylococci (CoNS) was determined using the cefoxitin (30μg) antibiotic disk diffusion method. *S. aureus* and CoNS were reported methicillin-resistant when the zone of inhibition was ≤21 and ≤24 mm while methicillin-sensitive when the zone of inhibition was ≥22 mm and ≥25 mm, respectively. Antibiotic discs gentamicin (10μg), amikacin (30μg), ceftazidime (30μg), tetracycline (30μg), trimethoprim/sulfamethoxazole (1.25/23.75μg),

ciprofloxacin (30μg), meropenem (10μg), imipenem (10μg), ampicillin (10μg), ceftriaxone (30μg), amoxicillin-clavulanic acid (20μg), and ciprofloxacin (30μg) (Oxoid Ltd and HiMEDIA LLC, Pennsylvania, USA) were used to assess the antimicrobial susceptibility pattern of Gram-negative bacteria. The zone of inhibition around the antimicrobial discs was measured to the nearest millimetre using a graduated calliper. Finally, the isolates were classified as sensitive, intermediate, and resistant to the tested drugs according to CLSI 2021 guideline [26].

## Data quality control

The questionnaire was prepared in English and translated into Amharic which was then translated back to English for consistency. The filled questionnaires were daily checked for completeness. Standardized procedures were used for specimen collection and the collected specimens were processed within 6 hours of collection after appropriate preservation. The quality of culture media and the expiry date of reagents was checked before performing each test. Culture media were prepared aseptically by autoclaving and 5% of each batch was checked for sterility through overnight incubation at 37˚C. Quality and performance of the culture media and antibiotics were also checked by inoculating standard bacterial strains of *S. aureus* ATCC® 25923, *E. coli* ATCC® 25922, *P. aeruginosa* ATCC® 27853, and *S. pneumoniae* ATCC® 49619.

## Data analysis and interpretations

Data were cleaned, coded, and entered EpiData version 4.2 software and exported to SPSS version 22 software for analysis. Descriptive statistics were used to summarize data while bivariate logistic regression was applied to determine the association between predictors and outcome variables with a 95% confidence interval. Variables with a *P*-value ≤0.25 in the crude analysis were subjected to adjusted analysis through multivariate logistic regression with a 95% confidence interval to control confounding factors. *P*-values ≤ 0.05 were considered statistically significant. Results were presented by using graphs and tables based on the type of data.

## Ethical consideration

The study was conducted after it was ethically approved by the Research and Ethical Review Committee of the College of Health Sciences, Debre Markos University (Protocol Number: DMU/CHS/RERC/65/2020). Written assent of care givers/guardians (for participants below 18 years old) and consent (for adults) were obtained before data collection. All the information obtained from the study participants was coded to keep confidentiality. Test results were communicated with the clinicians of the eye clinic for appropriate interventions.

## Operational definitions

**Ocular infection**: Eye infections occurring when harmful microorganisms, bacteria, fungi, and viruses invade any part of the eyeball or the surrounding area [27].

   **External ocular infection**: Eye infections occurring at the outer part of the eye including infectious diseases of the lids, conjunctiva, cornea, and lacrimal apparatus [28].

   **Prevalence of culture-confirmed bacteria**: This indicates the prevalence of bacteria isolated from EOI patients using the routine culture method. This bacterial prevalence represents only aerobic/facultative anaerobic bacteria and bacteria not requiring special media for growth.

   **Acute physical eye injury**: A physical eye injury that occurs within 24 hours [29].

**More frequent face washing**: Face washing two or more times per day with soap and water.

**Face washing once a day**: Face washing once a day with soap and water.

**Less frequent face washing**: Face washing occasionally less than once a day using soap and water.

**Multi-drug resistance (MDR)**: Bacteria that resist more than one drug in three or more classes of antimicrobial drugs [30].

**Systemic disease**: The diseases of the eye that directly or indirectly result from a disease process originating from another part of the body [31].

## Results

### Socio-demographic and clinical features of the study participants

Two hundred seven study participants clinically diagnosed with EOI were included in the study. Of the study participants, 57.5% (119/207) were males. The age of the study participants ranged from one year to 88 years with a median value of 59 years. The study participants in the age group of ≥ 65 years accounted for 37.7% (78/207). About 75.8% (157/207) of the study participants were rural residents, 50.2% (104/207) were farmers, 65.2% (135/207) were married and 76.3% (158/207) were illiterates (can't read and write) (**Table 1**).

Most patients with EOI were diagnosed with blepharitis and conjunctivitis with a respective prevalence of 46.9% (97/207) and 27.5% (57/207). Blepharitis was more common in males (58.4%) and in the age group of ≥65 years (38.6%) while dacryocystitis was more prevalent in females (85.7%) and in the age group of 15–24 years (57.1%). Hordeolum was reported only in the age group of 15–24 years, and it was more prevalent in females (66.7%). Besides,

**Table 1. Prevalence of culture-confirmed bacteria stratified by socio-demographic characteristics of the study participants at DMCSH, Northwest Ethiopia, 2021.**

| Variables | | Frequency, N (%) | Culture positive, N (%)$ |
|---|---|---|---|
| Sex | Male | 119(57.5) | 77(64.7) |
| | Female | 88(42.5) | 53(60.2) |
| Age in year | ≤14 | 12(5.8) | 5(41.7) |
| | 15–24 | 52(25.1) | 26(50) |
| | 25–64 | 67(32.4) | 42(62.7) |
| | ≥65 | 78(37.7) | 57(73) |
| Marital status | Married | 135(65.2) | 91(67.4.0) |
| | Single | 37(17.9) | 17(45.9) |
| | Widowed | 20(9.7) | 14(70) |
| | Divorced | 15(7.2) | 8(53.3) |
| Resident | Urban | 50(24.2) | 27(54) |
| | Rural | 157(75.8) | 103(65.6) |
| Educational status of participants | Read and write | 49(23.7) | 26(53) |
| | Not read and write | 158(76.3) | 104(65.8) |
| Occupations | Civil servant | 19(9.2) | 6(31.6) |
| | Farmer | 104(50.2) | 72(69.2) |
| | Merchant | 6(2.9) | 5(83.3) |
| | Housewife | 34(16.4) | 22(64.7) |
| | Daily laborer | 3(1.4) | 2(66.7) |
| | Others | 41(19.8) | 23(56) |

Key: $ = Proportion is calculated using the number of study participants in each category as a denominator.

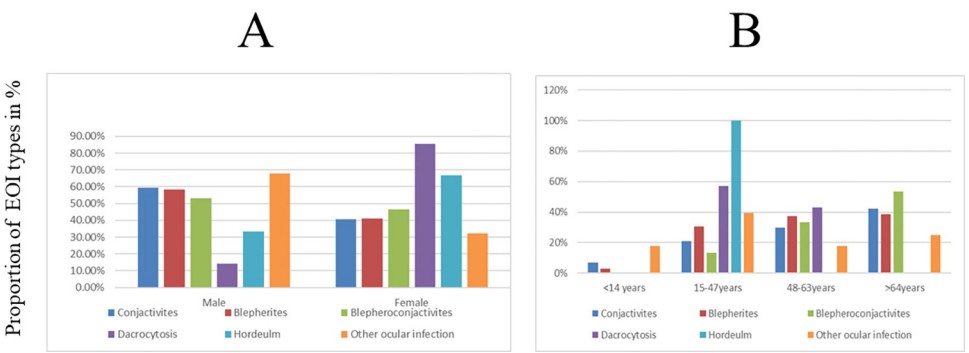

**Fig 1.** Distribution of external ocular infection among patients stratified by gender (A) and age group (B) at DMCSH, Northwest, Ethiopia.

conjunctivitis and blepharoconjunctivitis were more prevalent in males (59.6%, 53.3%) and in the age group of ≥65 years (42.1%, 53.3%). Other EOIs were more common in males (67.9%) and in the age group of 15–24 years (39.3%) (**Fig 1**).

## Prevalence of bacterial isolates

The overall prevalence of culture-confirmed bacterial isolates was 62.8% (130/207) with a 95% CI of 56.0–69%. Mixed bacterial isolates were not found in a single sample in this study. Among the isolates, 78.5% (102/130) were Gram-positive. *S. aureus* was the predominant Gram-positive bacteria accounting for 58.8% (60/102) followed by CoNS with a 26.5% (27/102) prevalence and *S. pneumoniae* with a proportion of 7.8% (8/102). From Gram-negative isolates, *E. coli* was the predominant bacterial isolate with a prevalence of 32.1% (9/28) followed by *P. mirabilis* (21.4%, 6/28). The least isolated bacteria were *S. pyogenes* from Gram positives (0.98%, 1/102) and *P. aeruginosa* from Gram negatives (3.6%, 1/28). The prevalence of culture-confirmed bacteria was 64.7% (77/119) in males, 73% (57/78) in age group of ≥65 years, 67.4% (91/135) in married, 65.6% (103/157) in rural residents, 65.8% (104/158) in those unable to read and write (illiterates), and 69.2% (72/104) in farmers (**Table 1**).

Most of the bacterial isolates were recovered from blepharitis (46.1%, 60/130) followed by conjunctivitis (30%, 39/130) and blepharoconjunctivitis (10%, 13/130) patients. Culture confirmed bacterial prevalence was the least among patients clinically diagnosed with hordeolum with a prevalence of only 2.3% (3/130). The predominant bacterial isolates observed in blepharitis cases were *S. aureus* (58.3%, 35/60) and CoNS (21.7%, 13/60). Among Gram-negative bacteria, *E. coli* was predominant in blepharitis patients (5%, 3/60) followed by *K. pneumoniae* (3.3%, 2/60) and *Enterobacter* species (3.3%, 2/60) (**Table 2**).

## Antimicrobial susceptibility patterns of bacterial isolates

The antimicrobial susceptibility pattern of Gram-positive bacterial isolates was tested on twelve antibiotics. A significant number of culture-confirmed bacterial isolates were resistant to one or more antimicrobial agents. Among the isolates, *S. aureus* showed high sensitivity to ciprofloxacin (83.3%, 50/60) followed by cefoxitin (80%, 48/60) and clindamycin (71.7%, 43/60). But *S. aureus* was highly resistant to azithromycin (63.3%, 38/60), penicillin (56.7%, 34/60), erythromycin (60.0%, 36/60) and doxycycline (45.0%, 27/60). Based on cefoxitin resistance, the prevalence of methicillin-resistant *S. aureus* (MRSA) was 20% (12/60). Besides, 25.9% (7/27) methicillin-resistant coagulase-negative staphylococci (MRCoNS) were identified. *S. pneumoniae* showed high sensitivity to clindamycin (87.5%, 7/8) but highly resistant to

**Table 2. Distribution of culture-confirmed bacterial isolates stratified by type of EOI at DMCSH, Northwest Ethiopia, 2021.**

| Isolates | Clinical diagnosis | | | | | | |
|---|---|---|---|---|---|---|---|
| | Conjunctivitis N = 57, n (%) | Belphroconjactivitus, N = 15, n (%) | Blepharitis N = 97, n (%) | Dacryocystitis N = 7, n (%) | Hordeolum N = 3, n (%) | Others N = 28, n (%) | Total N = 207, n (%) |
| **Gram-positive** | | | | | | | |
| *S. aureus* | 14(23.3) | 5(8.3) | 35(58.3) | 2(3.3) | 2(3.3) | 2(3.3) | 60(100) |
| CoNS | 9(33.3) | 2(7.4) | 13(48.1) | 1(3.7) | 0(0.0) | 2(7.4) | 27(100) |
| *S. pneumoniae* | 2(25) | 0(0.0) | 3(37.5) | 1(12.5) | 0(0.0) | 2(25) | 8(100) |
| *S. pyogenes* | 0(0.0) | 0(0.0) | 1(100) | 0(0.0) | 0(0.0) | 0(0.0) | 1(100) |
| *S. agalactiae* | 2(50) | 0(0.0) | 0(0.0) | 0(0.0) | 1(25) | 1(25) | 4(100) |
| Viridans streptococci | 1(50) | 1(50) | 0(0.0) | 0(0.0) | 0(0.0) | 0(0.0) | 2(100) |
| Sub total | 28(27.5) | 8(7.8) | 52(51.0) | 4(3.9) | 3(2.9) | 7(6.9) | 102(100) |
| **Gram-negative** | | | | | | | |
| *E. coli* | 5(55.5) | 1(11.1) | 3(33.3) | 0(0.0) | 0(0.0) | 0(0.0) | 9(100) |
| *P. aeruginosa* | 0(0.0) | 0(0.0) | 0(0.0) | 1(100) | 0(0.0) | 0(0.0) | 1(100) |
| *K. pneumoniae* | 2(40) | 1(20.0) | 2(40) | 0(0.0) | 0(0.0) | 0(0.0) | 5(100) |
| *P. mirabilis* | 2(33.3) | 1(16.7) | 1(16.7) | 0(0.0) | 0(0.0) | 2(33.3) | 6(100) |
| *Citrobacter* species | 0(0.0) | 2(66.7) | 0(0.0) | 0(0.0) | 0(0.0) | 1(33.3) | 3(100) |
| *Enterobacter* species | 2(50.0) | 0(0.0) | 2(50) | 0(0.0) | 0(0.0) | 0(0.0) | 4(100) |
| Sub total | 11(39.3) | 5(4.9) | 8(28.6) | 1(3.6) | 0(0.0) | 3(10.7) | 28(100) |
| Total | 39(30.0) | 13(10.0) | 60(46.2) | 5(3.8) | 3(2.3) | 10(7.7) | 130(100) |

CoNS = Coagulase-negative staphylococci.

penicillin and ceftazidime each accounting for 50% (4/8). *S. pyogenes*, *S. agalactiae*, and viridans streptococci were highly sensitive to clindamycin, gentamicin, and chloramphenicol; however, they exhibited resistance to trimethoprim/sulfamethoxazole, doxycycline, and ampicillin. Some Gram-positive bacteria showed intermediate sensitivity to antibiotics (**Table 3**).

Among Gram-negative bacterial isolates, *E. coli* showed a 100% sensitivity to ciprofloxacin, meropenem, and imipenem (9/9), and a high sensitivity to trimethoprim/sulfamethoxazole as well as amikacin 77.8% (7/9). On the other hand, *E. coli* demonstrated high resistance to amoxicillin-clavulanic acid and tetracycline each accounting for 44.4% (4/9), and ampicillin and ceftazidime each accounting for 33.3% (3/9).

*P. mirabilis* showed a 100% sensitivity to ciprofloxacin, amikacin, chloramphenicol, and meropenem. Additionally, this isolate also showed high susceptibility to trimethoprim/sulfamethoxazole 83.3% (5/6), gentamycin, ampicillin, and ceftazidime (each accounting for 66.7% (4/6)). However, *P. mirabilis* showed less sensitivity to amoxicillin-clavulanic acid, imipenem, and tetracycline (each accounting for 50% (3/6)). *P. aeruginosa*, *K. pneumoniae*, and *Citrobacter* isolates were 100% sensitive to amikacin, imipenem, and ciprofloxacin (**Table 4**).

## Multidrug-resistance patterns of bacterial isolates

Among the total culture-confirmed bacterial isolates (n = 130), 59.2% (77/130) of them demonstrated MDR pattern. Among Gram-positive isolates, 65.7% (67/102) were MDR while only 32.1% (9/28) of the Gram-negative isolates were found to be MDR. *S. aureus* (63.2%, 43/68) and CoNS (29.4%, 20/68) showed a high percentage of MDR pattern. From Gram-negative isolates, *E. coli* (55.5%, 5/9) and *Enterobacter* species (22.2%, 2/9) exhibited a high level of MDR pattern (**Table 5**).

**Table 3. Antibiotic susceptibility pattern of culture-confirmed Gram-positive bacteria isolated from EOI patients at DMCSH, Northwest Ethiopia, 2021.**

| Isolates (N = 130) | | Antimicrobial susceptibility pattern | | | | | | | | | | | |
|---|---|---|---|---|---|---|---|---|---|---|---|---|---|
| | | GM n(%) | Fox n(%) | P n(%) | C n(%) | SXT n(%) | AZM n(%) | CLD n(%) | CIP n(%) | CAZ n(%) | AMP n(%) | ERY n(%) | DOX n(%) |
| S. aureus | S | 40(66.7) | 48(80) | 18(30) | 40(66.7) | 39(65) | 17(28.3) | 43(71.7) | 50(83.3) | NT | NT | 15(25.0) | 24(40) |
| | I | 11(18.3) | - | 8(13.3) | 3(5) | 1(1.7) | 5(8.3) | 1(1.7) | 2(3.3) | NT | NT | 9(15.0) | 9(15) |
| | R | 9(15) | 12(20) | 34(56.7) | 17(28.3) | 20(33.3) | 38(63.3) | 16(26.7) | 9(15) | NT | NT | 36(60.0) | 27(45) |
| CoNS | S | 15(55.6) | 20(74.1) | 10(37) | 20(74.1) | 21(77.8) | 8(29.6) | 20(74.1) | 25(92.6) | NT | NT | 13(48.1) | 17(63) |
| | I | 6(22.2) | - | 3(11.1) | 4(14.8) | 3(11.1) | 4(14.8) | 2(7.4) | 1(3.7) | NT | NT | 10(37) | 3(11) |
| | R | 6(22.2) | 7(25.9) | 14(51.9) | 3(11.1) | 3(11.1) | 17(63) | 5(18.5) | 1(3.7) | NT | NT | 4(14.8) | 5(18.5) |
| S. pneumoniae | S | 3(37.5) | NT | 2(25) | 5(62.5) | 6(75) | 5(62.5) | 7(87.5) | 4(50) | 4(50) | 4(50) | 6(75) | 4(50) |
| | I | 2(25) | NT | 2(25) | 2(25.0) | 2(25) | 2(25) | 1(12,5) | 0(0.0) | 0(0.0) | 3(37.5) | 2(25) | 3(37.5) |
| | R | 3(37.5) | NT | 4(50) | 1(12.5) | 0(0.0) | 1(12.5) | 0(0.0) | 4(50.0) | 4(50) | 3(37.5) | 0(0.0) | 1(12.5) |
| S. pyogenes | S | 1(100) | NT | NT | 1(100) | NT | 1(100) | 1(100) | 1(100) | 1(100) | 1(100) | NT | 1(100) |
| | I | 0(0.0) | NT | NT | 0(0.0) | NT | 0(0.0) | 0(0.0) | 0(0.0) | 0(0.0) | 0(0.0) | NT | 0(0) |
| | R | 0(0.0) | NT | NT | 0(0.0) | NT | 0(0.0) | 0(0.0) | 0(0.0) | 0(0.0) | 0(0.0) | NT | 0(0) |
| S. agalactiae | S | 3(75) | 2(50) | 3(75) | 2(50) | 1(25) | 2(50) | 3(75) | 4(100) | NT | 2(50) | 1(25) | 2(50) |
| | I | 1(25) | 2(50) | 0(0.0) | 2(50) | 1(25) | 0(0.0) | 0(0.0) | 0(0.0) | NT | 1(25) | 1(25) | 0(0) |
| | R | 0(0.0) | 0(0.0) | 1(25) | 0(0.0) | 2(50) | 2(50) | 1(25) | 0(0.0) | NT | 1(25) | 2(50) | 2(50) |
| Viridans streptococci | S | 2(100) | 2(100) | 2(100) | 2(100) | NT | NT | 1(50) | 2(100) | 1(50) | 2(100) | 1(50) | 1(50) |
| | I | 0(0.0) | 0(0.0) | 0(0.0) | 0(0.0) | NT | NT | 0(0.0) | 0(0.0) | 0(0.0) | 0(0.0) | 1(50) | 1(50) |
| | R | 0(0.0) | 0(0.0) | 0(0.0) | 0(0.0) | NT | NT | 1(50) | 0(0.0) | 1(50) | 0(0.0) | 0(0.0) | 0(0.0) |

CoNS = Coagulase negative staphylococci, S = Sensitive, I = Intermediate, R = Resistance, AMC = Amoxicillin-clavulanic acid, AMP = Ampicillin, CIP = Ciprofloxacin, AK = Amikacin, C = Chloramphenicol, CLD = Clindamycin, TET = Tetracycline, SXT = Trimethoprim/sulfamethoxazole, ERY = Erythromycin, GM = Gentamicin, FOX = Cefoxitin, P = Penicillin, AZM = Azithromycin, CAZ = Ceftazidime, DOX = Doxycycline, NT = Not tested.

## Factors associated with culture-confirmed bacterial isolates

Different socio-demographic and clinical characteristics of the study participants were assessed for their possible association with the prevalence of culture-confirmed bacterial isolates in patients with EOI. In bivariate logistic regression, rural residence ($P = 0.141$), age group 15–24 years ($P = 0.030$), age group 25–64 years ($P = 0.043$), age groups $\geq 65$years ($P = 0.037$), merchant ($P = 0.136$), housewife ($P = 0.231$), farmer ($P = 0.082$), history of eye surgery ($P = 0.001$), frequency of face washing once a day ($P = 0.020$) and less frequency of face washing ($P = 0.010$) showed statistically significant association with culture-confirmed causes of EOI. After adjusting for confounding factors, age groups of 15–24 and 25–64 years, being farmer, history of eye surgery, less frequency of face washing, and face washing once a day were significant predictors of culture-confirmed bacterial causes of EOI ($P < 0.05$) (**Table 6**).

Patients in the age groups of 15–24 (AOR: 9.18, 95% CI: 1.01–82.80; $P = 0.049$) and 25–64 (AOR: 7.47, 95% CI: 1.06–52.31; $P = 0.043$) years had a statistically significant odds of harbouring culture-confirmed bacteria. On the other hand, patients who had a history of eye surgery were 5.39 times more likely to harbour culture-confirmed bacteria than their counterparts were (AOR: 5.39, 95% CL: 1.66–17.48; $P = 0.005$). Besides, the odds of harbouring culture-confirmed bacteria in patients who used to wash their face less frequently was 5.32 times (AOR: 5.33, 95% CI: 1.31–7.23; $P = 0.010$). In addition, patients washing their face once a day were 3.08 times more likely to be infected with culture-confirmed bacteria compared to patients who used to wash their face more frequently (AOR: 3.08, 95% CI: 1.13–25.13; $P = 0.035$) (**Table 6**).

**Table 4. Antibiotic susceptibility pattern of culture-confirmed Gram-negative bacterial isolates from EOI patients at DMCSH, Northwest Ethiopia, 2021.**

| Isolates (n = 130) | | Antimicrobial susceptibility pattern | | | | | | | | | | |
|---|---|---|---|---|---|---|---|---|---|---|---|---|
| | | GM N(%) | AMK N(%) | AMC N(%) | C N(%) | AMP N(%) | TET N(%) | SXT N(%) | MER N(%) | IPM N(%) | CIP N(%) | CAZ N(%) |
| *E. coli* | S | 3(33.3) | 7(77.8) | 5(55.6) | 6(66.7) | 4(44.4) | 4(44.4) | 7(77.8) | 9(100) | 9(100) | 8(88.9) | 4(44.4) |
| | I | 3(33.3) | 1(11.1) | 0(0.0) | 1(11.1) | 2(22.2) | 1(11.1) | 1(11.1) | 0(0.0) | 0(0.0) | 0(0.0) | 2(22.2 |
| | R | 2(22.2) | 1(11.1) | 4(44.4) | 2(22.2) | 3(33.3) | 4(44.4) | 1(11.1) | 0(0.0) | 0(0.0) | 1(11.1) | 3(33.3) |
| *P. aeruginosa* | S | 1(100) | 1(100) | NT | NT | NT | NT | NT | 1(100) | 1(100) | 1(100) | NT |
| | I | 0(0.0) | 0(0.0) | NT | NT | NT | NT | NT | 0(0.0) | 0(0.0) | 0(0.0) | NT |
| | R | 0(0.0) | 0(0.0) | NT | NT | NT | NT | NT | 0(0.0) | 0(0.0) | 0(0.0) | NT |
| *K. pneumoniae* | S | 3(60) | 5(100) | 2(40) | 4(80) | 1(20) | 3(60) | 4(80) | 4(80) | 5(100) | 5(100) | 5(100) |
| | I | 1(20) | 0(0.0) | 2(40) | 0.(0.0) | 2(40.0) | 0(0.0) | 1(20) | 1(20) | 0(0.0) | 0(0.0) | 0(0.0) |
| | R | 1(20) | 0(0.0) | 1(20) | 1(20) | 2(40) | 2(40) | 0(0.0) | 0(0.0) | 0(0.0) | 0(0.0) | 0(0.0) |
| *P. mirabilis* | S | 4(66.7) | 6(100) | 3(50) | 6(100) | 4(66.7) | 3(50) | 5(83.3) | 6(100) | 3(50) | 6(100) | 4(66.7) |
| | I | 1(16.7) | 0(0.0) | 1(16.7) | 0(0.0) | 0(0.0) | 1(16.7) | 0(0.0) | 0(0.0) | 3(50) | 0(0.0) | 2(33.3) |
| | R | 1(16.7) | 0(0.0) | 2(33.3) | 0(0.0) | 2(33.3) | 2(33.3) | 1(16.7) | 0(0.0) | 0(0.0) | 0(0.0) | 0(0.0) |
| *Citrobacter* species | S | 1(33.3) | 3(100) | 1(33.3) | 2(66.7) | 3(100) | 1(33.3) | 2(66.7) | 3(100) | 3(100) | 3(100) | 1(33.3) |
| | I | 2(66.7) | 0(0.0) | 1(33.3) | 1(33.3) | 0(0.0) | 1(33.3) | 1(33.3) | 0(0.0) | 0(0.0) | 0(0.0) | 1(33.3) |
| | R | 0(0.0) | 0(0.0) | 1(33.3) | 0(0.0) | 0(0.0) | 1(33.3) | 0(0.0) | 0(0.0) | 0(0.0) | 0(0.0) | 1(33.3) |
| *Enterobacter* species | S | 2(50) | 3(75) | 2(50) | 2(50) | 1(25) | 1(25) | 4(100) | 3(75) | 3(75) | 3(75) | 3(75) |
| | I | 1(25) | 0(0.0) | 1(25) | 1(25) | 1(25) | 2(50) | 0(0.0) | 0(0.0) | 0(0.0) | 1(25) | 1(25) |
| | R | 1(25) | 1(25) | 1(25) | 1(25) | 2(50) | 1(25) | 0(0.0) | 1(25) | 1(25) | 0(0.0) | 0(0.0) |

S = Sensitive, I = Intermediate, R = Resistance, AMC = Amoxicillin-clavulanic acid, AMP = Ampicillin, CIP = Ciprofloxacin, AK = Amikacin, C = Chloramphenicol, TET = Tetracycline, SXT = Trimethoprim/sulfamethoxazole, GM = Gentamicin, MER = Meropenem, IPM = Imipenem, CAZ = Ceftazidime, NT = Not tested.

**Table 5. Multidrug resistance patterns of culture-confirmed bacterial isolates from EOI patients at DMCSH, Northwest Ethiopia, 2021.**

| Isolates | Total | MDR patterns | | | | | |
|---|---|---|---|---|---|---|---|
| | | R0 N(%) | R1 N(%) | R2 N(%) | R3 N(%) | R4 N (%) | ≥R5 N(%) |
| *S. aureus* | 60 | 2(3.3) | 5(8.3) | 10(16.7) | 14(23.3) | 14(23.3) | 16(26.7) |
| CoNS | 27 | 2(7.4) | 2(7.4) | 5(18.5) | 8(29.6) | 8(29.6) | 3(11.1) |
| *S. pneumoniae* | 8 | 4(50) | 1(12.5) | 1(12.5) | 2(25.0) | 0(0.0) | 0(0.0) |
| *S. pyogenes* | 1 | 1(100) | 0(0.0) | 0(0.0) | 0(0.0) | 0(0.0) | 0(0.0) |
| *S. agalactiae* | 4 | 1(25) | 0(0.0) | 2(50) | 2(50) | 1(25) | 0(0.0) |
| Viridans streptococci | 2 | 2(100) | 0(0.0) | 0(0.0) | 0(0.0) | 0(0.0) | 0(0.0) |
| *E. coli* | 9 | 2(22.2) | 2(22.2) | 0(0.0) | 2(22.2) | 1(11.1) | 2(22.2) |
| *P. aeruginosa* | 1 | 0(0.0) | 1(100) | 0(0.0) | 0(0.0) | 0(0.0) | 0(0.0) |
| *K. pneumoniae* | 5 | 1(20) | 2(40) | 1(20.0) | 1(20.0) | 0(0.0) | 0(0.0) |
| *P. mirabilis* | 6 | 2(33.3) | 3(50) | 0(0.0) | 1(16.7) | 0(0.0) | 0(0.0) |
| *Citrobacter* species | 3 | 1(33.3) | 2(66.7) | 0(0.0) | 0(0.0) | 0(0.0) | 0(0.0) |
| *Enterobacter* species | 4 | 0(0.0) | 2(50) | 0(0.0) | 2(50.0) | 0(0.0) | 0(0.0) |
| Total | 130 | 18(13.8) | 20(15.4) | 19(14.6) | 32(24.6) | 24(18.5) | 2(1.5) |

CONS = Coagulase-negative staphylococci, Ro = bacterial isolate sensitive to all antibiotics, R1 = bacterial isolate resistance to one antibiotics, R2 = bacterial isolate resistance to two antibiotics of different classes, R3 = bacterial isolate resistance to three antibiotics of different classes, R4 = bacterial isolate resistance to four antibiotics of different classes, and >R5 = bacterial isolate resistance to five and above antibiotics of different classes.

**Table 6. Association of socio-demographic and clinical factors with bacterial isolates in EOI patients at DMCSH, Northwest Ethiopia, 2021.**

| Variables | | Isolates N (%)$ | COR (95%CI) | P-value | AOR (95%CI) | P-value |
|---|---|---|---|---|---|---|
| Sex | Female | 53(40.8) | 1 | | | |
| | Male | 77(59.2) | 1.22(0.697–2.131) | 0.487 | - | - |
| Age | ≤14 years | 5(3.8) | 1 | | 1 | |
| | 15–24 years | 26(20) | 4.05 (1.15–14.29) | 0.030* | 9.18(1.01–82.80) | 0.049** |
| | 25–64 years | 42(32.3) | 3.37(1.01–11.31) | 0.048* | 7.47(1.06–52.31) | 0.043** |
| | ≥65 years | 57(43.8) | 1.99(1.04–3.80) | 0.037* | 1.88(0.80–4.41) | 0.148 |
| Residence | Urban | 27(20.8) | 1 | | 1 | |
| | Rural | 103(79.2) | 1.62(0.85–3.101) | 0.141* | 0.92 (0.25–3.38) | 0.908 |
| Educational status | Read and write | 26(20.0) | 1 | | | |
| | Not read and write | 104(80.0) | 0.88(0.19–3.94) | 0.872 | - | - |
| Occupation | Civil servant | 6(4.6) | 1 | | 1 | |
| | Farmer | 72(55.4) | 2.77(0.88–8.72) | 0.082* | 5.33(1.04–37.328) | 0.045** |
| | Merchant | 5(3.8) | 0.57(0.27–1.19) | 0.136* | 0.37(0.09–1.52) | 0.169 |
| | Housewife | 22(16.9) | 0.25(0.027–2.40) | 0.231* | 0.19(0.01–3.83) | 0.283 |
| | Daily labourer | 2(1.5) | 0.69(0.27–1.77) | 0.449 | 0.47(0.09–2.54) | 0.385 |
| | Others | 23(17.7) | 0.64(0.05–7.62) | 0.723 | 1.08(0.06–19.18) | 0.955 |
| Use of traditional medicine | Yes | 5(3.8) | 0.33(.038–2.87) | 0.314 | - | - |
| | No | 125(96.2) | 1 | | | |
| History of eye trauma | Yes | 11(8.5) | 1.25(0.48–3.27) | 0.643 | - | - |
| | No | 119(91.5) | 1 | | | |
| History of eye surgery | Yes | 21(16.2) | 6.46 (2.18–19.16) | 0.001* | 5.39 (1.66–17.48) | 0.005** |
| | No | 109(83.8) | 1 | | 1 | |
| Frequency of face washing | Less frequent | 54(41.5) | 3.37(1.67–6.79) | 0.001* | 5.32 (1.31–7.23) | 0.010** |
| | Once a day | 69(53.1) | 4.74(1.45–15.47) | 0.020* | 3.07 (1.13–25.13) | 0.035** |
| | More frequent | 7(5.4) | 1 | | 1 | |
| Systemic disease | Yes | 5(3.8) | 0.69(0.12–3.89) | 0.889 | - | - |
| | No | 125(96.2) | 1 | | | |

Key

$ = Proportion is calculated using total bacterial isolate (130) as a denominator

*P-value <0.25

**P-value <0.05, COR-Crude Odds Ratio, AOR-Adjusted Odds Ratio, CI- Confidence Interval.

## Discussion

Out of the 207 study participants, 130 (62.8%, 95% CI: 56.0–69%) harboured culture-confirmed bacteria which is in line with previous studies conducted in different parts of Ethiopia (59.4%, 62.4%, 60%, 58.3%, 57.8%, and 60.8%) [1,8,16,24,32,33] and in India [17] where a 61% prevalence of bacteria-caused EOI was reported. However, this finding is lower than a study conducted in the Republic of Yemen (74.1%), Nigeria (74.9%), and Southwest Ethiopia (74.7%) [22,34,35], but it is higher than a study conducted in Bangalore (34.5%), Gondar (47.4%), Jimma (48.8%) and Addis Ababa (54.2%) [21,36–38]. This difference could be due to variations in geographical location, study design, and socioeconomic status of the populations.

In the current study, Gram-positive cocci were the most common culture-confirmed isolates with a prevalence of 77.7% (95% CI: 71.9–86) which is in line with several other studies from Ethiopia; Gondar (74.2%) [1], Addis Ababa (72.2%) [39], St. Paul Hospital Millennium Medical College, Addis Ababa (74.6%) [40] and with studies conducted abroad; Nigeria

(86.5%) [35], Italy (81.8%) [41] and India (79.69%) [17]. However, it is lower than a study conducted in Gondar 88% [16] and Borumeda Hospital 93.7% [8] but higher than reports from Jimma 52% [22], Hawassa 61.5% [3], Bahir Dar 66.3% [24], Shashemene 68.2% [33] and Egypt 58.9% [42]. Like these studies, Gram-positive cocci were the most common bacteria isolated from EOI patients [43–45] owing to the availability of these pathogens as commensals on the skin. Consistent with previous studies in Ethiopia at Gondar [16], Bahir Dar [24], Shashemene [33], Addis Ababa [39], and a recent study from Italy [41] the predominant Gram-positive cocci were *S. aureus* and CoNS. The increased prevalence of Gram-positive cocci may be due to contamination of the eye from skin normal flora as a result of touching eyes with hands, cataract extraction, and through contact lens [1,9]. However, the respective prevalence of *S. aureus (*46.1%) and CoNS (20.5%) in this study is higher compared to other studies conducted in Addis Ababa (36.8%) [39], Bahir Dar (37% and 23.1%) [24], Shashemene (37.4%) [33] and Yemen's Sana'a city (30.1% and 8.2%) [34]. But, this prevalence is lower than a study from Gondar (50.3% and 33.5%) [16] and Shashemene (28.8%, (in the case of CoNS) [33]. This difference could be due to differences in sample size where our sample size is comparatively smaller.

The prevalence of culture-confirmed Gram-negative bacteria in the current study was 21.5% which is in line with a study from Addis Ababa (18.8%) [40], but lower than studies from Bahir Dar (33.7%) [24], Gondar (44.5%) [36], Shashemene (31.8%) [33] and India (35%) [38]. However, it is higher than studies conducted in Borumeda, Gondar, and Jimma, which reported a respective prevalence of 6.5%, 12%, and 11.5% [8,16,22]. In our study, *E. coli* was the predominant Gram-negative bacteria similar to studies conducted in Shashemene, Ethiopia [33], Nigeria [35] and Italy [41]. On the contrary, other studies conducted in Jimma [22], Yemen [46], and Gondar [20] reported *P. aeruginosa* as the dominant Gram-negative bacterial isolate. Moreover, studies by Ayehubizu *et al.* [24] and Getahun *et al.* [16] reported *K. pneumoniae* as a predominant Gram-negative bacteria.

Unlike previous studies conducted in Ethiopia which reported the dominance of conjunctivitis [16,24,33,47], the commonest type of EOI in the current study was blepharitis with a prevalence of 46.2% followed by conjunctivitis (28.1%). Like the studies, *S. aureus* was the predominant isolate in blepharitis and conjunctivitis patients in the current study. However, the prevalence of *S. aureus* in blepharitis (58.3%) and conjunctivitis (35.9%) in the current study is higher compared with studies conducted in different parts of Ethiopia Menelik II referral hospital (37.8% and 22.6%) [37], Bahir Dar (15% and 26.6%) [24], Shashemene (47.2% and 30.8%) [33] and Tigray (38.5% and 21.5%) [47]. However, the prevalence of blepharitis and conjunctivitis in the current study is lower than a study from Gondar (50.6% and 51.1%) [16]. The high rate of *S. aureus* and CoNS among blepharitis cases may be associated with the presence of virulence factors such as exo-enzymes which have the potential of introduction by hands and aerosols [48].

Dacryocystitis causes irritation and discomfort to the patient which commonly affects the middle-aged and elderly women [49]. In this study, the prevalence of dacryocystitis was 3.6% predominantly caused by Gram-positive bacteria which is lower than other studies conducted in Ethiopia; Bahir Dar (45.5%) [24], Gondar (7.3%) [16], Shashemene (17.7%) [33] and Tigray (4.8%) [47]. Gram-positive bacteria predominantly cause dacryocystitis in this study similar to different studies in Ethiopia [16,24,33,47] and Iran [18]. On the contrary, a study from Israel [50] reported a predominance of Gram-negative bacteria in dacryocystitis patients. In this regard, an earlier study [49] also noted the increasing trend of Gram-negative bacteria especially in the case of chronic dacryocystitis. This scenario is different in different study settings. For example, a study [51] reported a high predominance of CoNS in chronic dacryocystitis while *S. aureus* and *Pseudomonas* species predominate in the case of acute dacryocystitis. The

cause of these epidemiological differences could be due to differences in bacterial ecology owing to regional and climate differences as reported in keratitis and conjunctivitis patients [52,53].

Prescription of antibiotics without conducting antimicrobial susceptibility testing for severe ocular infections is routinely practiced resulting in an increased rate of antimicrobial resistance [54]. In this study, culture-confirmed bacterial isolates showed a resistance pattern of 63.3%, 60%, 56.6%, 45.5%, and 44.4% to azithromycin, penicillin, erythromycin, tetracycline, and doxycycline, respectively. This is lower compared to other studies done in Ethiopia; Hawassa (69.9%) [3] and Gondar (77.1%) [36]. Most of the isolated bacteria were sensitive to ciprofloxacin which is in agreement with a study conducted in Addis Ababa [40]. *S. aureus* showed high resistance to azithromycin, penicillin, erythromycin, and doxycycline, which is partly similar to studies done in Bahir Dar, Gondar, and Shashemene [16,24,33] where the authors showed a high resistance pattern of *S. aureus*, especially to penicillin. The current study reported a high resistance pattern of *S. pneumoniae* to ceftazidime and penicillin. The high prevalence of penicillin-resistant *S. pneumoniae* in this study is contrary to a study from Shashemene [33] where the authors reported 100% susceptibility of *S. pneumoniae* to penicillin. On the other hand, CoNS showed high resistance to penicillin like previous studies from Ethiopia [16,24,33].

We observed a 20% prevalence of MRSA based on cefoxitin resistance. This result is high compared to studies conducted at St. Paul Hospital Millennium Medical College (12.5%) [40], Bahir Dar (16.9%) [24] and the United Kingdom (8.3%) [55] but lower than studies conducted in Menelik II Referral Hospital (34.3%) [37] Gondar (24%) and Uganda (31.9%) [9]. In our study, *S. aureus* was the predominant cause of culture-confirmed dacryocystitis which showed a high sensitivity to ciprofloxacin similar to studies from Ethiopia [47] and Israel [51].

In this study, *E. coli* was observed to be resistant to amoxicillin-clavulanic acid, tetracycline, ampicillin, and ceftazidime. *P. mirabilis* also showed less sensitivity to amoxicillin-clavulanic acid, imipenem, and tetracycline. Other studies also reported a high resistance profile of *E. coli* and *Proteus* species to tetracycline and ampicillin [16,24,33]. Other Gram-negative bacteria including *P. aeruginosa*, *K. pneumoniae, and Citrobacter* exhibited 100% sensitivity to amikacin, imipenem and ciprofloxacin in this study, which is similar to previous studies conducted in Ethiopia [16,24,33] where the authors reported high susceptibility of *P. aeruginosa*, *K. pneumoniae*, and *Citrobacter* especially to ciprofloxacin.

We observed a high prevalence of MDR among Gram-positive bacteria (66.7%) where *S. aureus* (63.2%), CoNS (29.4%), and *S. pneumoniae* (2.9%) showed a high multidrug resistance profile. Among Gram-negative bacteria, 32.1% were MDR in the present study. Overall, 59.2% of the culture-confirmed bacterial isolates were MDR. This is lower than previous studies conducted in Ethiopia; Gondar (64.6%) and St. Paul's Hospital Millennium Medical College (66.4%) [16,40], southern Ethiopia (69.9%) [54] and other studies from Gondar (87.1%) [1] and (77.3%) [36]. However, it is higher than studies from Tigray (53.9%) [11] and Bahir Dar (45.2%) [24].

Overall, the observed MDR pattern of culture-confirmed bacteria to different antibiotics could be linked to prescription of broad-spectrum antibiotics, lack of regular screening of antimicrobial resistance patterns before prescription, self-medication practice, and misuse of antibiotics [8,20,24]. The antimicrobial resistance pattern of bacteria causing ocular diseases differs from place to place and time to time due to different drug regulatory policies and bacterial ecology [56]. The differences in antimicrobial susceptibility patterns of bacterial isolates against different antimicrobials in different countries and/or settings might be due to the differences in bacterial strain, laboratory procedures, bacterial load, laboratory facility, drug control policies, and awareness of the community towards drug resistance.

In the present study, different socio-demographic and clinical variables were significant causes of bacterial EOI. The prevalence of culture-confirmed bacteria was significantly associated with age groups 15–24 years *(P = 0.049)* and 25–64 years (*P* = 0.043), being farmer (*P* = 0.045), previous history of eye surgery (*P* = 0.005*) and* washing face less frequently (*P* = 0.01) and washing face once a day (*P* = 0.035). This result is comparable with the data reported from Dessie [8]. In this study, although not a significant predictor (P = 0.148), most culture-confirmed bacteria were isolated from individuals in the age group ≥ 65 years old which might be due to age-associated deterioration of immunity supported by a study done in Menelik II referral hospital [37] and ALERT center, Ethiopia [39]. Children and elderly are known to be more susceptible to EOI [15] in support of our findings. A study conducted in ALERT center [39] reported that residence and educational status didn't show any statistically significant association with bacterial prevalence in EOI patients (P>0.05) which is similar to this study.

This study has the following limitations: 1) conducted in a single center 2) lack of facilities to isolate Chlamydia, anaerobic bacteria, and slow-growing bacteria 3) we did not report the vancomycin resistance pattern of *S. aureus* while it was the predominant isolate in the study 4) we didn't isolate fungi and molecular methods were not employed and 5) a short time frame and smaller sample size. Having these limitations, our study provides a glimpse into the incidence, antimicrobial susceptibility patterns and associated factors of culture-confirmed bacterial causes of EOI important for clinicians and policymakers to design appropriate interventions.

## Conclusion

The prevalence of culture-confirmed bacteria isolated from EOI patients was significantly high in the study area. The major bacterial isolates were *S. aureus*, CoNS, *S. pneumoniae*, *E. coli*, and *P. mirabilis*. Ciprofloxacin was a comparatively more effective antibiotic for both Gram-positive and Gram-negative bacteria. However, high rate of bacterial antibiotic resistance was observed in this study. In addition, a considerable proportion of Gram-positive and Gram-negative bacterial isolates demonstrated MDR. Young and old age (below 65 years), being a farmer, previous history of eye surgery and less frequency of face washing significantly increased the incidence of culture-confirmed bacterial causes of EOI. Strict guidelines and drug regulation policies should be in place for the prevention and control of antimicrobial resistance. Additionally, bacterial isolation and antimicrobial susceptibility testing should be routinely performed, and public health measures are also pivotal to tackling EOI caused by bacteria.

## Supporting information

**S1 Data. File 1: Data set included in the analysis.**
(SAV)

## Acknowledgments

We would like to acknowledge Debre Markos University and DMCSH for their administrative support. Further, we duly acknowledge DMCSH eye clinic and microbiology laboratory staffs for assisting in data collection and allowing the laboratory space. We duly acknowledge Arnaud John Kombe Kombe (PhD) for editing the language of the manuscript.

## Author Contributions

**Conceptualization:** Zewodie Haile, Hylemariam Mihiretie Mengist, Tebelay Dilnessa.

**Data curation:** Zewodie Haile, Hylemariam Mihiretie Mengist, Tebelay Dilnessa.

**Formal analysis:** Zewodie Haile.

**Funding acquisition:** Zewodie Haile.

**Investigation:** Zewodie Haile, Hylemariam Mihiretie Mengist, Tebelay Dilnessa.

**Methodology:** Zewodie Haile, Hylemariam Mihiretie Mengist, Tebelay Dilnessa.

**Project administration:** Zewodie Haile, Hylemariam Mihiretie Mengist, Tebelay Dilnessa.

**Resources:** Zewodie Haile, Hylemariam Mihiretie Mengist, Tebelay Dilnessa.

**Software:** Zewodie Haile, Hylemariam Mihiretie Mengist, Tebelay Dilnessa.

**Supervision:** Zewodie Haile, Hylemariam Mihiretie Mengist, Tebelay Dilnessa.

**Validation:** Zewodie Haile, Hylemariam Mihiretie Mengist, Tebelay Dilnessa.

**Visualization:** Zewodie Haile, Hylemariam Mihiretie Mengist.

**Writing – original draft:** Zewodie Haile.

**Writing – review & editing:** Zewodie Haile, Hylemariam Mihiretie Mengist, Tebelay Dilnessa.

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
