## [Decision Letter · Decision Letter 0]

26 Jul 2022

PONE-D-22-13636Bacterial Isolates, Antimicrobial Susceptibility Pattern and Associated Factors of External Ocular Infections among Patients attending Eye Clinic at Debre Markos Comprehensive Specialized Hospital, Northwest EthiopiaPLOS ONE

Dear Dr. Mengist,

Thank you for submitting your manuscript to PLOS ONE. After careful consideration, we feel that it has merit but does not fully meet PLOS ONE’s publication criteria as it currently stands. Therefore, we invite you to submit a revised version of the manuscript that addresses the points raised during the review process.

I have received the reviews of your manuscript. While your paper addresses an interesting question, the reviewers stated several concerns about your study and did not recommend publication in its present form. One reviewer felt that the rationale of the study needs to be strengthen, as well as the criteria of sample selection and collection. The other reviewers felt that the manuscript could benefit from substantial editing for clarity.  Please see the reviewers’ insightful comments below

We look forward to receiving your revised manuscript.

Kind regards,

Baochuan Lin, Ph.D.

Academic Editor

PLOS ONE

Journal Requirements:

Reviewers' comments:

Reviewer's Responses to Questions

**Comments to the Author**

1. Is the manuscript technically sound, and do the data support the conclusions?

Reviewer #1: Yes

Reviewer #2: Partly

Reviewer #3: Partly

2. Has the statistical analysis been performed appropriately and rigorously? 

Reviewer #1: Yes

Reviewer #2: No

Reviewer #3: Yes

3. Have the authors made all data underlying the findings in their manuscript fully available?

Reviewer #1: Yes

Reviewer #2: Yes

Reviewer #3: Yes

4. Is the manuscript presented in an intelligible fashion and written in standard English?

Reviewer #1: No

Reviewer #2: No

Reviewer #3: Yes

5. Review Comments to the Author

Reviewer #1: The aim of this study was to identify the prevalence of bacteria isolated from ocular samples and their pattern of resistance to antibiotics. This article is very interesting for readers of this journal. The language of the paper is difficult to understand and includes many errors, and the results must be better reported. The main text and the methods have to need to improve.

1) Improve the title, it is too long; also the keywords.

2) The abstract is too long. Some details of statistical data would be reported in the main text and not in the abstract. The introduction and result sections in abstract could be improved, and reduce Method section.

3) Improve the Introduction, to be more specific and clear, reference https://doi.org/10.3390/microorganisms8071033. Report epidemiological data that focus on the mobility of this disease in the world.

4) Table 3,4: could only show R% and no I% or S%, it is better.

5) In the discussion section, reference this article and compare it with your results, https://doi.org/10.3390/antibiotics11040463

Reviewer #2: Thank you very much for the opportunity to review the above-referenced manuscript titled "Bacterial Isolates, Antimicrobial Susceptibility Pattern and Associated Factors of External Ocular Infections among Patients attending Eye Clinic at Debre Markos Comprehensive Specialized Hospital, Northwest Ethiopia" by Haile et al. Considering the increasing burden of ocular bacterial infections and the concurrent variation in its etiology studies investigating into such pertinent issues are warranted. There is an existing similar published peer-reviewed manuscript from this same area of North-Western Ethiopia. However, Haile and colleagues provide contextual evidence, which I found elusive considering the lack of necessary details to determine the soundness of the study.

Major comments

1. What was the authors' rationale for selecting the Debre Markos Comprehensive Specialized Hospital?

2. Are their patients' characteristics different from that of Gondar teaching hospital regarding socioeconomic and lifestyle factors since all facilities are located in the same region and with similar published literature from the latter?

3. How representative is the sample?

4. Authors, in their limitation, mentioned the "lack of facilities to isolate fastidious bacteria, anaerobic bacteria, and slow-growing bacteria," which contradicts some elements of the methodology, and I require authors to address the below query.

5. What necessitated the authors to use chocolate agar (traditionally and widely known for the growth of fastidious organisms) for the initial isolation process?

6. Authors grow some organisms in the presence of 5% CO2; what was the rationale if there were facilities limitations.

7. Where chlamydia trachomatis investigated?

8. How was the sample transported, and with what transport medium?

9. Who collected the samples, and what was his/her expertise?

10. Who administered the questionnaire, and at what stage was it done?

Minor comments

1. Authors should present their hypothesis in the background to give meaning to the objective.

2. Could the authors provide some details of the facilities and the services they provide

3. The introduction could benefit from adequate structuring. Consider presenting in an inverted pyramid form, thus more broad and narrow it downwards.

4. By 'written assent', do the researchers mean taking written consent from caregivers? A more direct explanation will be ideal.

5. Consider deleting the operational terminologies since it appears redundant.

Reviewer #3: There is no author/s identified from the Department of Ophthalmology though this was given on the title page.

The author may state as a limitation those tests not carried out on fungi, molecular detection and keratitis cases.

6. PLOS authors have the option to publish the peer review history of their article (what does this mean?). If published, this will include your full peer review and any attached files.

Reviewer #1: No

Reviewer #2: No

Reviewer #3: No

---

## [Author Response · Author response to Decision Letter 0]

7 Aug 2022

Rebuttal letter

Manuscript number: PONE-D-22-13636

Title: Bacterial Isolates, Antimicrobial Susceptibility Pattern and Associated Factors of External Ocular Infections among Patients attending Eye Clinic at Debre Markos Comprehensive Specialized Hospital, Northwest Ethiopia, to be published in PLOS ONE

Dear Reviewers,

We really appreciate your effort made to make our manuscript readable through providing us with valuable comments and corrections. We have corrected all concerns you raised throughout the manuscript. We provided two types of documets; clean manuscript and another manuscript with track changes so that you can easily notice the revisions we made. Below is a detailed point-by-point response to your queries.

Point-by-point response

Result section 

I kindly recommend the authors rewrite this section, particularly the AST results. For example, there are texts read, as “Most Gram-positive and Gram-negative bacteria were sensitive to ciprofloxacin. However, 83.3% (85/102) of Gram-positive bacteria showed resistance to at least one drug.” Such kinds of generalizations are ambiguous for the reader and need to be specified and rewritten. 

1. Response: We corrected the statement and avoided vaue results.

Introduction 

Line 2: “Several factors including, but not limited to, dust, high temperature…” here the dust, temperature by them self may induce EOI? Need to clarify it.

2. Response: Corrected

Method section 

In this section, please include the Study participants and Socio-demographic data and please check to include the eligibility criteria in "... clinically diagnosed with external Eye infections and fulfilled the eligibility criteria during..."

You may revise the sentence as follows: ‘’Demographic data (age, sex, monthly income, educational level, occupation, and address and) and ophthalmic clinical data (history of repeated infections, duration of stay in the hospital, use of contact lenses, surgery, previous antibacterial therapy, systemic diseases, and use of traditional medicine) were collected using structured questionnaires and physical examinations, respectively.’’

3. Response: Thank you. We revised it accordingly.

The authors should include specimen collection, Transportation and processing of parts

It did not show specimen handling for external ocular infections on the method part.

4. Response: The specimen was collected at DMCSH and processed at DMCSH microbiology laboratory. Thus, no need to use transport media as the sample was directly processed in the laboratory. 

Antimicrobial susecepility testing 

“The methicillin resistance pattern of S. aureus and coagulase-negative staphylococci (CoNS) was determined using the cefoxitin (30μg) antibiotic disk diffusion method. S. aureus and CoNS were reported methicillin-resistant…” in this sentence cefoxitin was used as a surrogate test for CoNS. Is it recommended by the CLSI guidelines? I am in doubt please check out it. 

Cotrimoxazole and Trimethoprim / Sulfamethoxazole are two names for a single drug and it would be better two use one of the names in this or a single document.

5. Response: Yes using cefoxitin for determining methicillin resistance is recommended by CLSI and studies aalso recommend this. This is because Cefoxitin is considered as a better inducer of mec -A gene expression than oxacillin or methicillin (https://www.sciencedirect.com/science/article/pii/S0255085721017485). 

Results: 

 This section did not cover some important aspects of the study. For example, Keratitis is not covered in this manuscript as well as fungal infections. 

6. Response: Since its prevalence was very small, keratitis was reported together in “Other external ocular infections”. Fungal infections are not the objectives of the study and should not be reported.

Antimicrobial susceptibility patterns of bacterial isolates

The following drugs were not stated in the method section but simply appeared in the result section and were tested for some gram-positive isolates. Clindamycin. Doxycycline, trimethoprim, and sulfamethoxazole.

7. Response: Thank you. We included clindamycin and doxycycline in the methods. But trimethoprim-sulfamethoxazole was already included in its other name; cotrimoxazole.

The same kind of error has been introduced in gram-negative isolates. ciprofloxacin, meropenem, imipenem, ampicillin were not listed as drugs used for AST. However, there were in the result section. This indicated that your result is not based on your plan or May the result has been taken from other lab results. It should be based on what you had written in the method sections.

8. Response: We used all the drugs indicated in the results, however, we didn’t include in the methods section due to technical error; otherwise, we all understand scientific misbehaving. It is totally wrong to consider we took the results from other studies while we all are reputed researchers and senior medical microbiologists. Research s found on trust and we don’t encourage such kinds of statements. However, if a researcher is found with evidence using manulpulated data in his/her research, we encourage to claim retraction of papers.

Multidrug-resistance patterns of bacterial isolates

I recommend the author illustrate the class of antibiotics tested in table 5. In addition to that, the study did not mention the drug resistance genes. It needs to have a justification why not carried out those isolates, which are MDR.

9. Response: The classes of drugs are already discusses in tables 3 and 4. Readers are advised to have a first information from these tables. Including drug classes in Table 5 will be just a repetition of the previous results. So we just reported the summaries in Table 5.

Figures: The figure1 is unclear, together with the legend. Thus, it is suggested to improve them. The two variables (SEX and AGE) should be depicted separately against the clinical cases. 

10. Response: Thank you. We did it accordingly and the figure is now clear.

DISCUSSIONS 

In discussion parts... Is 62.9% in lined with 57.4%? You must have a 95% CI to justify the difference.

In this section, the author more focuses on the clinical importance of the isolates and AST pattern rather than reporting and comparing the magnitudes in a different location. Interpret and correlate the findings with their clinical relevance. 

11. Response: Our discussion is written well as prevalence studies better be copared and justified with the results other similar studies. There is no 62.9% result in our manuscript rather 62.8% (95% CI: 56.0-69%) which is in line with 57.4% as clearly indiated in the first paragraph of the discussion section with 95% CI.

Reviewer #1: The aim of this study was to identify the prevalence of bacteria isolated from ocular samples and their pattern of resistance to antibiotics. This article is very interesting for readers of this journal. The language of the paper is difficult to understand and includes many errors, and the results must be better reported. The main text and the methods have to need to improve.

1. Response: We tried to avoid grammatical errors and typos across the manuscript. 

1) Improve the title, it is too long; also the keywords.

 2. Response: Titles should explain atleast what and when. Besides, all specific objectives are advised to be indicated in the title. In this regard, we couldn’t do modifications here.

2) The abstract is too long. Some details of statistical data would be reported in the main text and not in the abstract. The introduction and result sections in abstract could be improved, and reduce Method section.

 3. Response: Thank you. We removed some sentences from the abstract.

3) Improve the Introduction, to be more specific and clear, reference https://doi.org/10.3390/microorganisms8071033. Report epidemiological data that focus on the mobility of this disease in the world.

 4. Response: We cited this reference to include more epidemiological data.

4) Table 3,4: could only show R% and no I% or S%, it is better.

 5. Response: Some drugs should be reported with R, I and S. To do so, we prepared the tables like that. If it was only binary data like yes/no, it is advisable to use only one i.e. either yes or no as the other can be known by deducting it from the total proportion. 

5) In the discussion section, reference this article and compare it with your results, https://doi.org/10.3390/antibiotics11040463

 6. Response: Thank you for sharing this recent study. We duly included it in the discussion happily. 

Reviewer #2: Thank you very much for the opportunity to review the above-referenced manuscript titled "Bacterial Isolates, Antimicrobial Susceptibility Pattern and Associated Factors of External Ocular Infections among Patients attending Eye Clinic at Debre Markos Comprehensive Specialized Hospital, Northwest Ethiopia" by Haile et al. Considering the increasing burden of ocular bacterial infections and the concurrent variation in its etiology studies investigating into such pertinent issues are warranted. There is an existing similar published peer-reviewed manuscript from this same area of North-Western Ethiopia. However, Haile and colleagues provide contextual evidence, which I found elusive considering the lack of necessary details to determine the soundness of the study.

 1. Response: Sure, there are similar studies in northwest Ethiopia. But, northwest Ethiopia is too vast including the three zones of Gojja, the four zones of Gondar and the Wolkait Humera zones. There are no similar studies in east gojjam zone so far, and e did the research to fill these gaps. Different research have different ne insights and we do have our own insights which has made the manuscript novel in this regard. Prevalence studies indicate a one time data which require repeated investigation. Therefore, it is not dscouragable to work on similar themes in different place, time and population.

Major comments

1. What was the authors' rationale for selecting the Debre Markos Comprehensive Specialized Hospital?

 2. Response: DMCSH was selected because it is the only rwfderral hospital with independent eye clinic. 

2. Are their patients' characteristics different from that of Gondar teaching hospital regarding socioeconomic and lifestyle factors since all facilities are located in the same region and with similar published literature from the latter?

 3. Response: There may not be significant lifestyle factors differences among residents of nowrthwestern Ethiopia. However, socioeconomic data is different even among similar town residents. Such kinds of studies are recommended to be conducted at each district even below zonal level because cross-sectional studies are one time events that should be repeated in place and time. 

3. How representative is the sample?

 4. Response: We utilized consecutive sampling technique which is considered as the best among non-probability sampling techniques. We used scientific methods for calculating the sample size and thus we beleiv it is representative enough. In principle, 10% of a population is considered as representative where ours is above 20%.

4. Authors, in their limitation, mentioned the "lack of facilities to isolate fastidious bacteria, anaerobic bacteria, and slow-growing bacteria," which contradicts some elements of the methodology, and I require authors to address the below query.

 5. Response: Thank you. We didn’t isolate some fastidious bacteria (Chlamydia), anaerobic bacteria and slow growing bacteria. The sentence is just to mean Chlamydia and we made corrections here.

5. What necessitated the authors to use chocolate agar (traditionally and widely known for the growth of fastidious organisms) for the initial isolation process?

 6. Response: We used blood agar and chocolate agrar as general enriched media to grow all types of bacteria. Then, Macconkey agar and MSA were used to selectively grow Gram-negative bacteria and Staphylococci, respectively. 

6. Authors grow some organisms in the presence of 5% CO2; what was the rationale if there were facilities limitations.

 7. Response: The presence of 5% CO2 is important for the growth of microaerophilic bacteria, not anerobic bacteria. We used candle jar for providing 5% CO2 for microaerophilic bacteria including Streptococci. Ths facilities for anaerobic bacteria and microaerophilic bacteria are quite different.

7. Where chlamydia trachomatis investigated?

 8. Response: We didn’t report C. trachomatis in this study.

8. How was the sample transported, and with what transport medium?

 9. Response: Since we did the lab procedures inside DMCSH, no transport media was used. We directly inoculated the samples in agars.

9. Who collected the samples, and what was his/her expertise?

 10. Response: As clearly indicated in the data collection part now, data and specimen were collected by optometrist nurses

10. Who administered the questionnaire, and at what stage was it done?

 11. Response: Optometrist nurses administered the questionnaire and done after specimen collection.

Minor comments

1. Authors should present their hypothesis in the background to give meaning to the objective.

2. Could the authors provide some details of the facilities and the services they provide

3. The introduction could benefit from adequate structuring. Consider presenting in an inverted pyramid form, thus more broad and narrow it downwards.

4. By 'written assent', do the researchers mean taking written consent from caregivers? A more direct explanation will be ideal.

5. Consider deleting the operational terminologies since it appears redundant.

 12. Response: We provided a hypothesis in the background. Regarding the facilities, we included what eye care related facilities are given by DMCSH. We included only eye care related facilities and services as our objective is related to eye infections. The introduction is from general to specific and we believe it is not necessary to restructure. We corrected the ethical issues in the manuscript and made it clear. Operational terminologies are very important in case of our manuscript. It helps readers understand the scenario. The opertational terms here are not general, rather they are terms we operationally used in this study. 

Reviewer #3: There is no author/s identified from the Department of Ophthalmology though this was given on the title page.

The author may state as a limitation those tests not carried out on fungi, molecular detection and keratitis cases.

1. Response: we included limitations on failure of identifying fungi and lack of using molecular methods. But keratitis is included in the “other ocular infections” part as it is a very small proportion to be independently reported. Ophtalmologists identified patients during the study. For this, we acknowledged them.

---

## [Decision Letter · Decision Letter 1]

19 Sep 2022

PONE-D-22-13636R1Bacterial isolates, their antimicrobial susceptibility pattern, and associated factors of external ocular infections among patients attending eye clinic at Debre Markos Comprehensive Specialized Hospital, Northwest EthiopiaPLOS ONE

Dear Dr. Mengist,

Thank you for submitting your manuscript to PLOS ONE. After careful consideration, we feel that it has merit but does not fully meet PLOS ONE’s publication criteria as it currently stands. Therefore, we invite you to submit a revised version of the manuscript that addresses the points raised during the review process.

All reviewers agreed that the revised version showed improvement, however, there are a few points that still need to be addressed. Please discuss the limitation of the study and justification of study done in a single center.  Also, one of the reviewers has concern that the previous comments were not well addressed (see reviewer's insightful comments below).

We look forward to receiving your revised manuscript.

Kind regards,

Baochuan Lin, Ph.D.

Academic Editor

PLOS ONE

Journal Requirements:

Reviewers' comments:

Reviewer's Responses to Questions

**Comments to the Author**

1. If the authors have adequately addressed your comments raised in a previous round of review and you feel that this manuscript is now acceptable for publication, you may indicate that here to bypass the “Comments to the Author” section, enter your conflict of interest statement in the “Confidential to Editor” section, and submit your "Accept" recommendation.

Reviewer #1: All comments have been addressed

Reviewer #2: (No Response)

Reviewer #3: All comments have been addressed

2. Is the manuscript technically sound, and do the data support the conclusions?

Reviewer #1: Yes

Reviewer #2: No

Reviewer #3: Yes

3. Has the statistical analysis been performed appropriately and rigorously? 

Reviewer #1: Yes

Reviewer #2: No

Reviewer #3: Yes

4. Have the authors made all data underlying the findings in their manuscript fully available?

Reviewer #1: Yes

Reviewer #2: Yes

Reviewer #3: Yes

5. Is the manuscript presented in an intelligible fashion and written in standard English?

Reviewer #1: No

Reviewer #2: Yes

Reviewer #3: Yes

6. Review Comments to the Author

Reviewer #1: The authors have addressed review comments raised in a previous round of review and the manuscript was improved and is now it is acceptable for publication.

Reviewer #2: (No Response)

Reviewer #3: Some comments are not well addressed. For instance, cotrimoxazole is not the same drug as either trimethoprim, or sulfamethoxazole. It is the combination of two drugs.

Please state the limitations of this study.

7. PLOS authors have the option to publish the peer review history of their article (what does this mean?). If published, this will include your full peer review and any attached files.

Reviewer #1: No

Reviewer #2: No

Reviewer #3: No

---

## [Author Response · Author response to Decision Letter 1]

20 Sep 2022

Rebuttal letter

Manuscript number: PONE-D-22-13636R1

Title: Bacterial Isolates, their Antimicrobial Susceptibility Pattern and Associated Factors of External Ocular Infections among Patients attending Eye Clinic at Debre Markos Comprehensive Specialized Hospital, Northwest Ethiopia

Dear Editor and reviewers,

Thank you so much for your time and comments. We have corrected all concerns you raised throughout the manuscript. We provided two types of documents; clean manuscript and another manuscript with track changes so that you can easily notice the revisions we made. Below is a detailed point-by-point response to your queries.

Point-by-point response

1. Discuss the justification of study done in a single center

Response: We included the justification in the methods section as follows

The study was conducted among EOI suspected patients at DMCSH, which is found in Debre Markos town, the capital of East Gojjam zone in Amhara National Regional State, Northwest Ethiopia. DMCSH is the only tertiary hospital providing health care services for over four million inhabitants of East Gojjam and West Gojjam zones and the surrounding areas. In addition, it is the only hospital with an independent tertiary eye clinic that provides both outpatient and inpatient services. All cases requiring tertiary care service in the area are referred to DMCSH. Besides, it has also a high patient flow as the eye clinic provides medical service for an average of 21,000 patients per year of which about 4,151 of them are clinically diagnosed as EOI. Moreover, it is the only hospital in the area providing bacterial culture and antimicrobial susceptibility testing services. Due to these reasons, DMCSH was selected as the only study site.

2. Cotrimoxazole is not the same drug as either trimethoprim, or sulfamethoxazole. It is the combination of two drugs. 

Response: Yes, we understand. We used a combination of trimethoprim/sulfamethoxazole in this study, and we previously called it as “cotrimoxazole”. For convenience we removed the name “cotrimoxazole” and replaced it with its standard naming Trimethoprim/sulfamethoxazole (SXT). We used this combination with a total dose of 25μg and with a respective trimethoprim/sulfamethoxazole dose of (1.25/23.75μg). This is the standard use based on CLSI guideline. 

3. Please state the limitations of this study.

Response: We stated the limitations of the study in the last paragraph of the Discussion section as follows;

This study has the following limitations: 1) conducted in a single center 2) lack of facilities to isolate Chlamydia, anaerobic bacteria, and slow-growing bacteria 3) we did not report the vancomycin resistance pattern of S. aureus while it was the predominant isolate in the study 4) we didn’t isolate fungi and molecular methods were not employed and 5) a short time frame and smaller sample size. Having these limitations, our study provides a glimpse into the incidence, antimicrobial susceptibility patterns and associated factors of culture-confirmed bacterial causes of EOI important for clinicians and policymakers to design appropriate interventions.

---

## [Decision Letter · Decision Letter 2]

24 Oct 2022

Bacterial isolates, their antimicrobial susceptibility pattern, and associated factors of external ocular infections among patients attending eye clinic at Debre Markos Comprehensive Specialized Hospital, Northwest Ethiopia

PONE-D-22-13636R2

Dear Dr. Mengist,

We’re pleased to inform you that your manuscript has been judged scientifically suitable for publication and will be formally accepted for publication once it meets all outstanding technical requirements.

Kind regards,

Ivone Vaz-Moreira, PhD

Academic Editor

PLOS ONE

Additional Editor Comments (optional):

Reviewers' comments:

Reviewer's Responses to Questions

**Comments to the Author**

1. If the authors have adequately addressed your comments raised in a previous round of review and you feel that this manuscript is now acceptable for publication, you may indicate that here to bypass the “Comments to the Author” section, enter your conflict of interest statement in the “Confidential to Editor” section, and submit your "Accept" recommendation.

Reviewer #1: All comments have been addressed

Reviewer #3: All comments have been addressed

2. Is the manuscript technically sound, and do the data support the conclusions?

Reviewer #1: Yes

Reviewer #3: Yes

3. Has the statistical analysis been performed appropriately and rigorously? 

Reviewer #1: Yes

Reviewer #3: Yes

4. Have the authors made all data underlying the findings in their manuscript fully available?

Reviewer #1: Yes

Reviewer #3: Yes

5. Is the manuscript presented in an intelligible fashion and written in standard English?

Reviewer #1: Yes

Reviewer #3: Yes

6. Review Comments to the Author

Reviewer #1: (No Response)

Reviewer #3: The authors addressed all the comments. Therefore, no more comments provided. I recommend the authors to proceed the next step.

7. PLOS authors have the option to publish the peer review history of their article (what does this mean?). If published, this will include your full peer review and any attached files.

Reviewer #1: No

Reviewer #3: No

---

## [Editor Report · Acceptance letter]

26 Oct 2022

PONE-D-22-13636R2 

Bacterial isolates, their antimicrobial susceptibility pattern, and associated factors of external ocular infections among patients attending eye clinic at Debre Markos Comprehensive Specialized Hospital, Northwest Ethiopia 

Dear Dr. Mengist:

I'm pleased to inform you that your manuscript has been deemed suitable for publication in PLOS ONE. Congratulations! Your manuscript is now with our production department. 

Kind regards, 

on behalf of

Dr. Ivone Vaz-Moreira 

Academic Editor

PLOS ONE